# Kinetochore dynein is sufficient to biorient chromosomes and remodel the outer kinetochore

Bram Prevo [1,2] ✉, Dhanya K. Cheerambathur[1], William C. Earnshaw [1] & Arshad Desai [2,3,4] ✉

Multiple microtubule-directed activities concentrate on mitotic chromosomes to ensure their faithful segregation. These include couplers and dynamics regulators localized at the kinetochore, the microtubule interface built on centromeric chromatin, as well as motor proteins recruited to kinetochores and chromatin. Here, we describe an in vivo approach in the *C. elegans* one-cell embryo in which removal of the major microtubule-directed activities on mitotic chromosomes is compared to the selective presence of individual activities. Our approach reveals that the kinetochore dynein module, comprised of cytoplasmic dynein and its kinetochore-specific adapters, is sufficient to biorient chromosomes; by contrast, this module is unable to support congression. In coordination with orientation, the dynein module directs removal of outermost kinetochore components, including dynein itself, independently of the other microtubule-directed activities and kinetochore-localized protein phosphatase 1. These observations indicate that the kinetochore dynein module is sufficient to biorient chromosomes and to direct remodeling of the outer kinetochore in a microtubule attachment state-sensitive manner.

During mitosis, multiple microtubule-directed activities localize to chromosomes in order to direct chromosome alignment and biorientation on the spindle. The primary microtubule-directed activities on chromosomes are force-generating motor proteins, localized to either kinetochores (cytoplasmic dynein, CENP-E) or to mitotic chromatin (chromokinesins), and the indirect force-generating Ndc80 module, which couples to dynamic microtubules to harness their polymerization dynamics[1–3]. The Ndc80 module in metazoans is comprised of the microtubule-binding Ndc80 and Ska complexes, whose cooperation is important for ordered transitions in end-coupled kinetochore-microtubule attachment stability that ensure accurate segregation[4–9]. As Ska complex recruitment and actions at the kinetochore depend on Ndc80[5,6,9,10], we refer to their coordinated action as that of the Ndc80 module. Other microtubule dynamics regulators concentrated at the kinetochore-microtubule interface and important

for proper chromosome segregation include kinesin-13 depolymerases, CLASPs, XMAP215 family proteins, and EB family plus-end tracking proteins[2,11]. Many of this latter group of factors act globally on microtubules, thus complicating analysis of their specific functions at kinetochores.

Cytoplasmic dynein and the Ndc80 module are both recruited to kinetochores by specific cofactors. Dynein is recruited via the Rod/Zwilch/Zw10 (RZZ) complex and the activating adapter Spindly, and the Ndc80 module via the kinetochore linker & scaffold Mis12 and Knl1 complexes[3,12]. In many species, including vertebrates, the Ndc80 module is additionally recruited by CENP-T[13–15]. CENP-E, a plus-end directed motor, is present at kinetochores of vertebrates as well as some invertebrate species; in human cells, CENP-E contributes to chromosome congression[16] and dynein recruitment[17]. In contrast to these kinetochore-localized microtubule-targeted

[1]Wellcome Centre for Cell Biology, University of Edinburgh, Edinburgh, UK. [2]Ludwig Institute for Cancer Research, La Jolla, CA, USA. [3]Department of Cell and Developmental Biology, School of Biological Sciences, University of California San Diego, La Jolla, CA, USA. [4]Department of Cellular and Molecular Medicine, University of California San Diego, La Jolla, CA, USA. ✉e-mail: bram.prevo@ed.ac.uk; abdesai@ucsd.edu

activities, chromokinesins are broadly recruited to mitotic chromatin, potentially via direct binding to DNA and via interaction with condensin complexes[1,18–22].

Prior phenotypic, biochemical, biophysical, and structural work on conserved chromosome segregation machinery has revealed a central role for the Ndc80 module in the formation of load-bearing end-coupled kinetochore-microtubule attachments[2,3], for the kinetochore dynein module in lateral capture and remodeling/stripping of the outermost corona region of the kinetochore following formation of end-coupled attachments[12,23], and for chromokinesins in moving chromosomes toward the spindle equator[1]. While these efforts have significantly advanced understanding of the molecular mechanisms underlying chromosome segregation, the complexity of the coordinated action of the multiple factors involved has limited understanding of their individual contributions in an in vivo context. We therefore decided to first characterize the effect of removing all major force generators, creating a blank slate on mitotic chromosomes with respect to microtubule interactions. We then compared the dynamics of blank slate chromosomes to engineered states where individual force generators were selectively present on mitotic chromosomes. We conducted this analysis in the early *C. elegans* embryo, where conserved chromosome segregation factors have been extensively investigated, the kinetochore assembly hierarchy in vivo is well-characterized, kinetochore composition is relatively streamlined (e.g., the CENP-T branch of Ndc80 module recruitment and the CENP-E motor are absent), and the diffuse line-shaped kinetochores of the holocentric chromosomes enable rapid and dynamic readout of chromosome orientation on the spindle. In this model, the Ndc80 module is required to form load-beading kinetochore-microtubule attachments and, in its absence, there is extensive chromosome missegregation[24,25]. The kinetochore dynein module accelerates the formation of load-bearing attachments and, in its absence, there is modest chromosome missegregation[26,27]. The major KIF4 family chromokinesin KLP-19 is important to congress and orient chromosomes, and in its absence, there is extensive chromosome missegregation[19].

Here, by employing an in vivo reconstruction approach, we show that the kinetochore dynein module is sufficient to both orient chromosomes and to remodel the outer kinetochore. The Ndc80 module and chromokinesin act in parallel to the kinetochore dynein module to orient and congress chromosomes; their dominant role explains why the orientation function of the kinetochore dynein module was not previously recognized. We speculate that a direct orientation role of kinetochore dynein may contribute to explaining the chromosome missegregation observed across metazoans when the kinetochore dynein module is perturbed.

## Results

### A microtubule-interaction blank slate on mitotic chromosomes

In the *C. elegans* embryo, the three major microtubule-directed activities on chromosomes are the Ndc80 module, the kinetochore dynein module, and the KIF4 family chromokinesin KLP-19, which is the major mitotic chromatin-localized motor in this system (Fig. 1a)[19,25]. Imaging of in situ GFP-tagged NDC80, DHC-1 (dynein heavy chain), and KLP-19 showed that, while the Ndc80 complex assembled onto kinetochores prior to nuclear envelope breakdown (NEBD), dynein was recruited to kinetochores only after NEBD (Fig. 1b; Supplementary Fig. 1). KLP-19 was present on chromosomes at NEBD and remains chromosome-associated through metaphase. Both KLP-19 and DHC-1 were removed from chromosomes in anaphase; by contrast, NDC80 persisted on the kinetochores of anaphase chromosomes. In this experimental model, the Ska complex is recruited significantly later than NEBD, and its recruitment requires Ndc80 complex engagement with the microtubule lattice[5].

To generate a blank slate with respect to microtubule interactions on mitotic chromosomes, we co-depleted KNL-1, which is essential for outer kinetochore assembly in *C. elegans*, and KLP-19. In wild-type embryos, the 12 chromosomes from the two pronuclei aligned and congressed rapidly, with anaphase onset occurring ~3 min after NEBD (Fig, 1c; Supplementary Movie 1). By contrast, blank slate chromosomes remained dispersed after NEBD on the spindle and became oriented parallel to the spindle pole-to-pole axis (Fig. 1c; Supplementary Movie 2). To quantify chromosome dynamics, we employed two measures: a bounding box that measures the width of the dispersion of all chromosomes on the spindle[5] (Fig. 1d), and the angular orientation of chromosomes relative to the spindle pole-to-pole axis at the time of NEBD and anaphase onset (Fig. 1e). The bounding box analysis indicated that for the blank slate, there was no significant congression of chromosomes toward the spindle equator (Fig. 1d). The angular orientation analysis indicated that, in contrast to the perpendicular orientation at the time of anaphase onset in controls, the majority of chromosomes exhibited parallel orientation relative to the spindle axis in the blank slate (Fig. 1e). Thus, in the absence of chromatin and kinetochore-localized force generators/couplers, mitotic chromosomes are distributed throughout the spindle and have their axes oriented parallel to the spindle axis. This nematic alignment, where the longitudinal axes of chromosomes and microtubules are in parallel orientation, is probably driven by the action of dynamic microtubule polymers pushing on chromosomes as passive objects and their subsequent confinement.

### The dynein module is sufficient to orient chromosomes

The generation and characterization of a blank slate with respect to mitotic chromosome-microtubule interactions enabled us to next engineer in vivo states in which only one of the major force generators is present on mitotic chromosomes (Fig. 2a). In all cases, we monitored chromosome distribution on the spindle and axial orientation of chromosomes relative to the spindle axis, and compared the outcomes to the blank slate. To create a chromokinesin-only state, we depleted KNL-1 ("ChrKin only"); to create an Ndc80 module-only state, we co-depleted KLP-19 and ROD-1, a subunit of the RZZ complex that recruits dynein to kinetochores ("Ndc80 module only"); to create a kinetochore dynein module-only state, we co-depleted KLP-19 and NDC80 in a strain harboring an RNAi-resistant transgene expressing a mutant form of NDC80 that disrupts the ability of its conserved calponin homology (CH) domain to dock onto the microtubule surface[5,28] ("Kt Dynein module only") (Fig. 2a). The use of this mutant form of NDC80, which has been characterized in prior work[5,26], minimized impact on overall kinetochore structure. For technical reasons related to strain construction, in select experiments, we could not have the transgene-encoded NDC80 CH domain mutant present. We, therefore, also analyzed the KLP-19 and NDC80 co-depletion on its own as a second kinetochore dynein module-only state (Supplementary Fig. 2a).

In the presence of only the chromokinesin KLP-19, which is equivalent to the previously characterized "kinetochore null" phenotype[25], the chromosomes from the oocyte and sperm nuclei moved to the spindle center and formed two clusters lacking any discernable orientation. No anaphase segregation was observed (Fig. 2a; Supplementary Movie 3). Thus, chromokinesin activity is sufficient to support a form of pseudo-congression, where the chromosomes from each pronucleus move toward the spindle equator and are tightly clustered together but are not bioriented and fail to segregate (Fig. 2a, b). In the presence of the Ndc80 module only (kinetochore dynein and chromokinesin absent), chromosomes exhibited delayed partial congression and extensive mis-orientation, leading to the formation of chromatin bridges during anaphase segregation (Fig. 2a, b; Supplementary Movie 4). Detailed analysis of the chromosome distribution revealed delayed partial congression supported by the Ndc80 module (Fig. 2b). In the wild type, this may reflect engagement of the Ndc80 module after initial chromokinesin-driven chromosome movement toward the spindle equator.

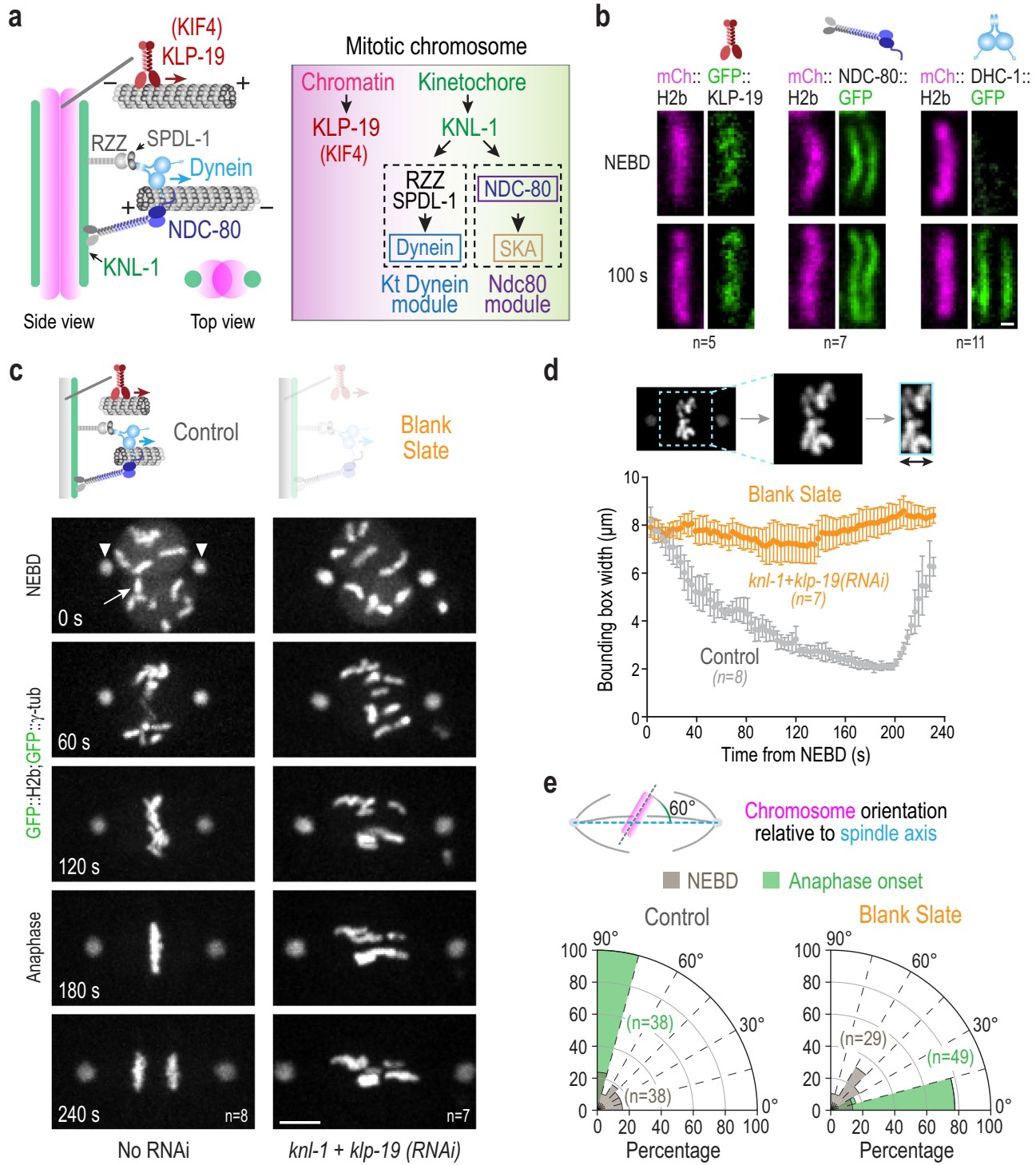

**Fig. 1 | Creating a blank slate on mitotic chromosomes with respect to spindle microtubule interactions in the *C. elegans* embryo. a** Schematics of the three major microtubule-targeting factors on mitotic chromosomes (left) and their assembly dependencies (right) in the one-cell *C. elegans* embryo. **b** Chromosomal localization of in situ-tagged GFP fusions of the indicated components on single chromosomes (mCherry::H2b) at NEBD and 100 s after NEBD. Scale bar, 0.5 μm. *n* is the number of embryos analyzed. **c** Phenotype of the blank slate generated by removing the chromokinesin KLP-19 and preventing outer kinetochore assembly by depletion of KNL-1. GFP fusions of histone H2b and γ-tubulin label the chromosomes (arrow) and spindle poles (arrowheads), respectively. Scale bar, 5 μm. *n* is

the number of embryos analyzed. **d** Minimal bounding box analysis quantifying chromosome dispersion on the spindle. Graph plots the mean bounding box width following NEBD for the indicated conditions. Error bars are the 95% confidence interval of the mean (CIM). *n* is the number of embryos analyzed. **e** Quantification of chromosome orientation relative to the spindle axis at NEBD and anaphase onset. Radial plots show the percentage of chromosomes within 15° angular orientation bins. Anaphase onset in the blank slate, where chromosome segregation fails, was scored by the initiation of spindle rocking. *n* represents the number of chromosomes measured per condition. Source data are provided as a Source Data file.

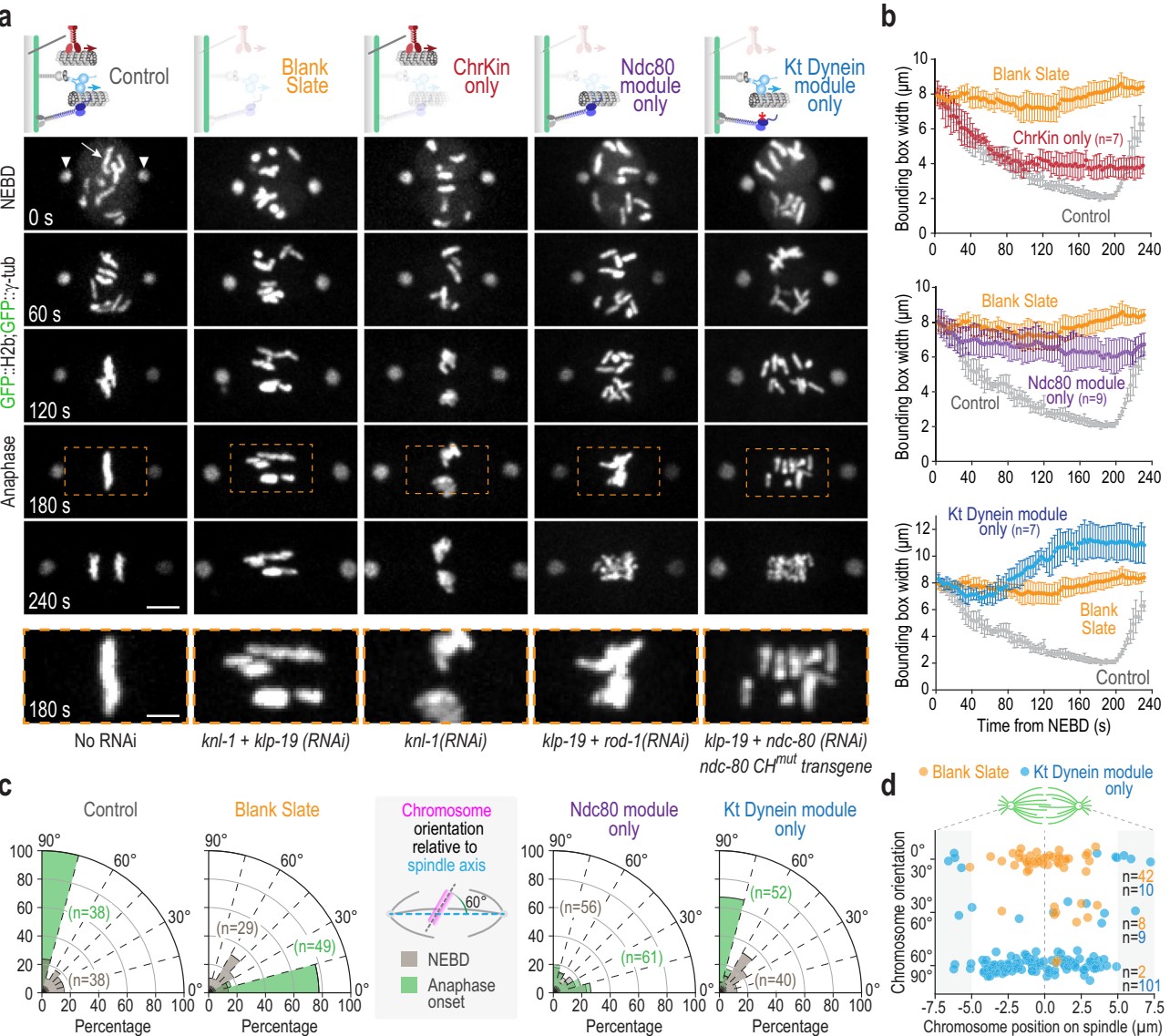

**Fig. 2 | An in vivo approach to define the contributions of individual microtubule-directed activities on mitotic chromosomes. a** Image panels from timelapse movies monitoring chromosomes (arrow) and spindle poles (arrowheads) for the indicated conditions. Time relative to NEBD is indicated in the lower left of the Control panels. Schematics (top) indicate the specific states created and analyzed. Dashed orange box in 180 s panels indicates the spindle region magnified below each set of timelapse panels. Labels below each column indicate the perturbation(s) used to create specific states. Scale bar, 5 μm (full spindle view), and 2.5 μm (magnified region). **b** Quantification of chromosome dispersion on the spindle performed as in Fig. 1d. The Control and Blank Slate curves are the same as in Fig. 1d and are plotted along with curves for specific perturbations to aid comparison. *n* is the number of embryos analyzed. Error bars are the 95% CIM. **c** Quantification of chromosome orientation relative to the spindle axis, similar to Fig. 1e. *n* is the number of chromosomes analyzed per condition. The Control and Blank Slate graphs are the same as in Fig. 1e and are plotted along with graphs for specific perturbations to aid comparison. **d** Plot of chromosome orientation, measured as in Fig. 1e, relative to chromosome position on the spindle for the blank slate and kinetochore dynein module-only states. The spindle equator is the origin of the x-axis and is indicated with a dashed line. The shaded areas indicate the 2.5 μm adjacent to the spindle pole. *n* is the number of chromosomes measured. Source data are provided as a Source Data file.

The most striking chromosome behavior observed was in the kinetochore dynein module-only states, generated either with or without the NDC80 microtubule-binding mutant present (Fig. 2a and Supplementary Fig. 2a; Supplementary Movie 5). As in the blank slate, chromosomes remained dispersed on the spindle when only the kinetochore dynein module was present (Fig. 2a, b), indicating that the kinetochore dynein module is unable to support congression. At the time of NEBD, in both the blank slate and when only the kinetochore dynein module was present, chromosomes were oriented randomly with respect to the spindle axis (Fig. 2c, gray wedges). However, in contrast to the blank slate where chromosomes became aligned parallel to the spindle axis, in the kinetochore dynein module-only state

the majority of chromosomes oriented perpendicular to the spindle axis (Fig. 2a, c, d and Supplementary Fig. 2a; Supplementary Movie 5). Thus, the kinetochore dynein module can orient chromosomes independently of chromokinesin and Ndc80 module activity. Plotting the angular orientation of chromosomes relative to their position on the spindle indicated the presence of a small number of pole-proximal chromosomes that failed to orient in the kinetochore dynein module-only state and remained aligned parallel to the spindle axis (Fig. 2d). A potential explanation for the parallel orientation of these polar chromosomes is that, in the absence of chromokinesin activity, minus end-directed kinetochore dynein motor activity traps them in the high-density parallel-oriented microtubule environment close to the

spindle poles. The presence of these persistent pole-associated chromosomes accounts for the greater average chromosome dispersion on the spindle in the kinetochore dynein module-only state relative to the blank slate (Fig. 2b). Taken together, these results highlight a potential direct role for the kinetochore dynein module in chromosome orientation on the spindle. A comparison of the Ndc80 module-only and the kinetochore dynein module-only states is informative (Fig. 2a–d and Supplementary Fig. 2a). The Ndc80 module drives late, partial congression but is unable to ensure proper chromosome orientation. By contrast, the kinetochore dynein module is unable to drive any congression but is remarkably efficient at orienting chromosomes.

### Dynein motor complex is required for chromosome orientation

The kinetochore dynein module is comprised of the RZZ complex, the kinetochore dynein adapter Spindly, and the dynein/dynactin motor complex. To assess whether the striking effect on chromosome angular orientation was indeed due to the action of kinetochore-localized dynein motor activity, we depleted ROD-1 under the conditions employed to generate the kinetochore dynein module-only state. Co-depletion of ROD-1 resulted in chromosomes exhibiting the parallel alignment observed for the blank slate (Fig. 3a and Supplementary Fig. 2b). While this result supported a role for RZZ complex-dependent dynein recruitment to kinetochores in driving chromosome orientation, it did not exclude the possibility that the RZZ complex and/or the kinetochore dynein activator Spindly (SPDL-1 in *C. elegans*) recruited by RZZ might act at kinetochores to orient chromosomes, independent of the dynein motor complex. We therefore next analyzed a point mutant in the conserved Spindly motif (where Phe199 is mutated to Ala), which does not affect RZZ or SPDL-1 recruitment but prevents dynein recruitment[26,29]. Comparing conditions where embryos expressed either WT or F199 > A SPDL-1 and were co-depleted of KLP-19 and NDC80 revealed that, while WT SPDL-1 supported the perpendicular alignment of chromosomes, the F199 > A SPDL-1 motif mutant caused chromosomes to align parallel to the spindle axis, as observed in the blank slate (Fig. 3b). Collectively, these results indicate that kinetochore dynein recruited in a RZZ and SPDL-1-dependent manner is sufficient to drive the striking perpendicular orientation of chromosomes relative to the spindle axis. In support of this conclusion, removal of dynein from microtubule plus ends by deletion of the plus-end-binding protein EBP-2[30] did not impact chromosome orientation when only the kinetochore dynein module was present (Supplementary Fig. 2c). We note that chromosome missegregation is observed when the kinetochore dynein module is perturbed across metazoans[27,31,32]. In *C. elegans* and *Drosophila*, inactivation of the spindle checkpoint does not explain the missegregation observed following RZZ complex inhibition[33–35] (Supplementary Fig. 3a–f). Thus, based on the results above, we suggest that a direct role of the kinetochore dynein module in chromosome orientation contributes to the missegregation observed across metazoans in the absence of kinetochore dynein function.

### Chromosomes biorient in the kinetochore dynein-only state

The striking change in chromosome orientation on the spindle observed in the kinetochore dynein-only state, despite chromosomes remaining dispersed on the spindle, suggested that kinetochore dynein might be sufficient not only for chromosome angular orientation relative to the spindle axis but also for biorientation, the state in which sister chromatids attach to opposite spindle poles. To assess if this was indeed the case, we imaged in situ GFP-tagged KNL-1, which marks the sister kinetochores on individual chromosomes as paired lines. Imaging of KNL-1::GFP revealed that, in the kinetochore dynein-only state, the majority of sister kinetochore pairs were indeed bioriented and faced opposite spindle poles despite their dispersion on the spindle (Fig. 3c). To assess the fate of these dispersed, bioriented chromosomes as the embryo progressed into anaphase, we imaged

KNL-1::GFP and GFP::H2b fusions. With both fusions, sister chromatids of perpendicularly oriented chromosomes were observed separating toward opposite spindle poles (Fig. 3d). We note that anaphase chromosome segregation in the *C. elegans* embryo is driven by spindle elongation[36]. As spindles prematurely elongate when load-bearing Ndc80 module-dependent kinetochore-microtubule attachments are absent[5,24], normal anaphase separation is not expected in the kinetochore dynein-only state. Rather, the partial separation of sisters observed for the perpendicularly oriented chromosomes provides additional evidence for the biorientation of sister kinetochores.

### Kinetochore dynein removal is coupled with orientation

A well-established function of the kinetochore dynein module is to remodel the outermost regions of the kinetochore following the formation of end-coupled attachments. Specifically, minus end motility of kinetochore dynein removes the spindle checkpoint-activating Mad1-Mad2 complexes in order to silence checkpoint signal generation and promote cell cycle progression[37,38]. During this process, a majority of kinetochore dynein is itself removed along with checkpoint components, with puncta of kinetochore dynein module components and associated checkpoint proteins observed moving toward spindle poles[37,39]. The precise mechanisms triggering this removal process are unclear aside from the observation that full removal and checkpoint silencing require end-on kinetochore-microtubule attachment[40]. One model is that Ndc80 complex engagement with microtubule ends is a pre-requisite to trigger dynein-dependent removal (Ndc80 engagement model). An argument against this model is that dynein-dependent removal of checkpoint proteins occurs despite Ndc80 depletion in human cells by RNAi[41]. A second model is that kinetochore-attached microtubules deliver a regulatory phosphatase activity that triggers dissociation between elements of the kinetochore dynein module and/or between the module and its binding interface on the outer kinetochore (Phosphatase delivery model[42]). To test these models and to gain additional insight into the biorientation function of the kinetochore dynein module, we imaged in situ GFP-tagged dynein heavy chain (DHC-1::GFP), in the kinetochore dynein module-only state. A snapshot ~100 s after NEBD revealed a heterogeneous population of chromosomes, all with different positions and orientations (Fig. 4a), which confirms orientation being a chromosome-autonomous process. The amount of DHC-1 on kinetochores varied widely between individual chromosomes and largely correlated with chromosome orientation. Sister kinetochores of laterally oriented chromosomes were enriched for DHC-1, whereas perpendicularly oriented chromosomes had DHC-1 concentrated on one or neither of their sister kinetochores (Fig. 4a).

To capture the detailed temporal relationship between chromosome orientation and dynein dynamics at the kinetochore, we imaged chromosomes and DHC-1::GFP at high temporal resolution. A single chromosome that exhibited all of the distinct phases of DHC-1 dynamics and achieved biorientation is shown in Fig. 4b. At NEBD, both sister kinetochores on this chromosome lacked DHC-1 but then began to recruit it simultaneously. The chromosome was oriented parallel to the spindle axis at this time. Soon after, the chromosome rotated while translocating toward the right spindle pole. Strikingly, this coincided with an abrupt loss of DHC-1 signal from the kinetochore facing that pole (Fig. 4b; Supplementary Movie 6). During orientation, poleward translocation, and DHC-1 removal from the right kinetochore, there was no reduction in DHC-1 levels on the left sister kinetochore (Fig. 4b); interestingly, DHC-1 removal was asymmetric, proceeding from the top of the right kinetochore (Supplementary Fig. 3g). Thus, orientation and dynein removal are coordinated kinetochore-autonomous events that reflect the engagement of one kinetochore with microtubules emanating from one spindle pole. Next, the left sister kinetochore was captured by the left pole, causing leftward translocation, and

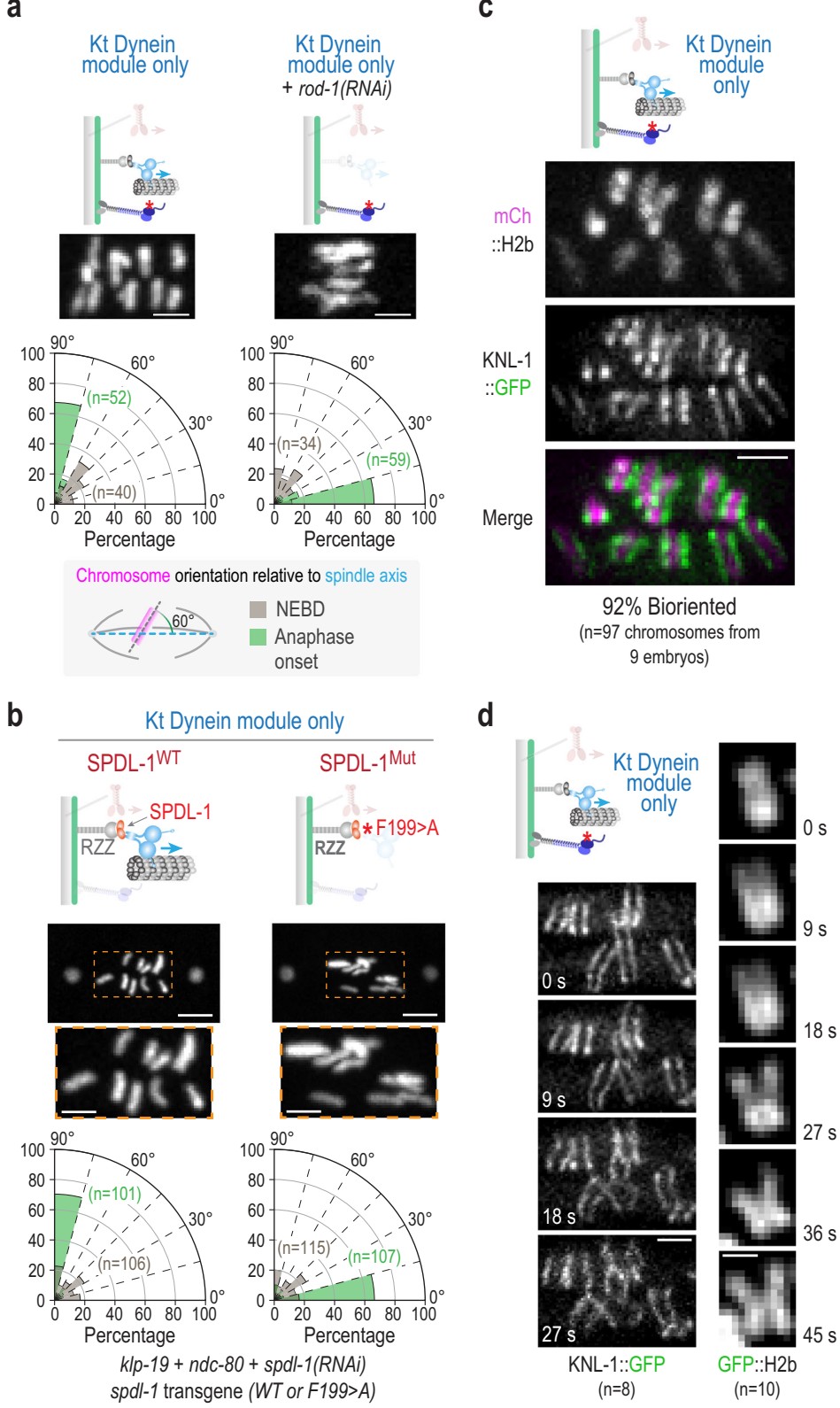

leading to the eventual loss of a large proportion of DHC-1 signal from this kinetochore (Fig. 4b).

Quantification of the amount of dynein and chromosome angle relative to the spindle pole-to-pole axis (Fig. 4b; graph) revealed; (1) the similar and continuous buildup of DHC-1 on both kinetochores during the initial phase, (2) the abrupt loss of most DHC-1 on the right sister kinetochore coincident with orientation of the chromosome perpendicular to the spindle axis, (3) subsequent capture and loss of DHC-1 from the left sister kinetochore, and (4) a residual pool of DHC-1 at the kinetochore that likely supports kinetochore-microtubule interactions in the bioriented state (Fig. 4b). Data for 9 kinetochores monitored in the kinetochore dynein-only state are shown in Fig. 4c. They all indicate coupling of orientation perpendicular to the spindle axis with DHC-1 removal. Orientation and DHC-1 removal also occurred

**Fig. 3 | Dynein recruitment by kinetochore-specific adapters is required to orient chromosomes in the kinetochore dynein module-only state.**
**a** Consequences of removal of the RZZ complex subunit ROD-1 in the condition used to generate the kinetochore dynein module-only state. Representative images with graphs plotting chromosome angles relative to the spindle axis, measured as in Fig. 1e, are shown. The kinetochore dynein module-only graph is the same as in Fig. 2c to aid comparison. Scale bars, 2.5 μm. **b** Comparison of SPDL-1 WT to the SPDL-1 mutant (F199 > A in the conserved Spindly motif) that perturbs dynein recruitment in the kinetochore dynein module-only state. Cartoons above depict the compared conditions; note that there was no transgene-encoded CH^mut NDC80

present. Representative images and graphs plotting chromosome angles relative to the spindle axis, measured as in Fig. 1e, are shown. Scale bars are 5 μm (full spindle view) and 2.5 μm (magnified region). **c** Images of in situ-tagged KNL-1::GFP and mCherry::H2b in the kinetochore dynein module-only state highlighting biorientation. Scale bar, 2 μm. **d** Images of in situ-tagged KNL-1::GFP and GFP::H2b in the kinetochore dynein module-only state highlighting poleward sister separation, despite lack of congression. The first frame was arbitrarily set to 0 s. Scale bars, 2 μm and 1 μm (KNL-1::GFP and GFP::H2b time series, respectively). *n* is the number of embryos analyzed. Source data are provided as a Source Data file.

coincidently with the translocation of the chromosome toward the engaged spindle pole (Fig. 4b; Supplementary Fig. 3h, i). In contrast to chromosomes that achieved perpendicular orientation, rare chromosomes that were trapped in a lateral orientation (Fig. 2d) maintained robust DHC-1 signal at both sister kinetochores until anaphase onset (Fig. 4d). DHC-1 dissociated from these chromosomes soon after anaphase onset. We hypothesize that loss at this time point is triggered by a global change in the cell cycle state rather than a specific microtubule attachment configuration.

In addition to DHC-1, we imaged the RZZ complex subunit ROD-1 and the kinetochore dynein activator SPDL-1 (Spindly). SPDL-1 behaved similarly to DHC-1 and was significantly depleted from oriented chromosomes (Fig. 5a and Supplementary Fig. 4a, b). In contrast, ROD-1 behaved differently and persisted on oriented chromosomes (Fig. 5a and Supplementary Fig. 4c). These observations suggest that the removal event primarily involves dissociation of the dynein motor complex and SPDL-1 from the RZZ complex, along with any SPDL-1-associated cargo.

A well-known cargo that is removed by the kinetochore dynein module is the spindle checkpoint-activating Mad1-Mad2 complex. We therefore also imaged in situ GFP-tagged MAD-1 in the kinetochore dynein-only state. Qualitatively, the behavior of MAD-1 was similar to that of DHC-1 and SPDL-1 (Supplementary Fig. 4d). Unfortunately, the high abundance of MAD-1 in the spindle area, along with its significantly slower recruitment to kinetochores relative to DHC-1, prevented us from quantifying its kinetochore dynamics (quantitative analysis of MAD-1 localization in *C. elegans* embryos requires the generation of monopolar spindles[43]). Overall, these results indicate that chromosome orientation and removal of the outermost kinetochore components such as DHC-1 and SPDL-1 are tightly coordinated in the kinetochore dynein module-only state.

### Dynein removal is independent of Ndc80 and kinetochore PP1

We next focused on addressing the mechanism that triggers dynein removal from kinetochores that is coupled to chromosome orientation. As we inactivated the NDC80 complex's microtubule-binding activity through mutation or depletion in the kinetochore dynein module-only state, our data are consistent with microtubule engagement by Ndc80 not being a cue for initiating dynein removal[41]. In support of this conclusion, comparison of the NDC80 microtubule-binding mutant to NDC80 WT, effectively comparing kinetochore dynein module-only to kinetochore dynein module plus NDC80 module states, revealed no significant difference in DHC-1 removal from kinetochores following chromosome orientation (Fig. 5b). In this analysis, we measured DHC-1 signal at both sister kinetochores immediately before and after one sister oriented toward a proximal pole. In addition to indicating robust removal of DHC-1 regardless of NDC80 functional status, this analysis confirmed that removal of DHC-1 is kinetochore-autonomous (Fig. 5b). The interval between the before and after measurement timepoints, which reflects the rate of removal, was also unaffected (Supplementary Fig. 5a). These data are consistent with significant Ndc80 depletion not preventing dynein-mediated removal of spindle checkpoint components in human cells[41].

We next tested whether kinetochore-localized phosphatase activity was important for triggering DHC-1 removal. In the *C. elegans* embryo, protein phosphatase 1 (PP1) docked onto KNL-1 is the primary kinetochore-localized phosphatase[44]. Unlike in vertebrates, where PP2A-B56 also concentrates at kinetochores, in situ GFP-tagged B56 subunits in *C. elegans* do not localize to kinetochores[44]; similarly, PP4 does not localize to kinetochores[45]. We thus focused on the delivery of protein phosphatase 1 (PP1) to its binding site on KNL-1 as a potential trigger for DHC-1 removal. PP1 localization to kinetochores occurs late after NEBD[44] and, in human cells, is inversely correlated with the localization of checkpoint proteins[46], suggesting that it could function as a removal trigger. To test this idea, we combined the kinetochore dynein module-only state with manipulation of KNL-1, the primary PP1-targeting protein at kinetochores in the *C. elegans* embryo[44,47]. Comparison of WT KNL-1 to a well-characterized PP1 docking mutant[44,47] (PP1^mut) revealed no significant difference in DHC-1 removal between these two conditions (Fig. 5b and Supplementary Fig. 5a), even though the PP1 docking mutant exhibited phenotypes such as delayed anaphase onset that were expected from prior work[44,47] (Supplementary Fig. 5b). The PP1 docking mutant of KNL-1 also did not affect removal of DHC-1 from the kinetochores of rare, persistently laterally attached chromosomes after anaphase onset (Supplementary Fig. 5c–e). These results argue against a model in which the formation of microtubule attachments delivers PP1 to trigger the release of dynein and associated cargo from the kinetochore. These data leave open the possibility that another regulatory activity initiates the removal of the kinetochore dynein module and associated cargo, with the proviso that any such activity must be sensitive to orientation/attachment state. Alternatively, removal may be intrinsic to the kinetochore dynein module once it achieves a specific attachment state, potentially an end-coupled state (see discussion below).

## Discussion

Here, we describe an experimental approach in the *C. elegans* embryo in which the presence of individual microtubule-targeting activities on mitotic chromosomes was compared to a blank slate, where all major microtubule-directed activities were absent from chromosomes. This approach revealed that the kinetochore dynein module is sufficient to biorient chromosomes on the spindle and remodel the outer kinetochore. Both of these actions occur in a chromosome-autonomous and kinetochore-autonomous manner. Removal of kinetochore dynein by RZZ complex inhibition has been associated with chromosome missegregation in *Drosophila*, *C. elegans*, and human cells[27,31,32], with the *Drosophila* observations being over three decades old. However, the reasons for this missegregation have remained unclear. At least in *C. elegans* and *Drosophila*, where the loss of the spindle checkpoint does not result in penetrant missegregation[33–35], perturbation of the spindle checkpoint cannot explain the missegregation observed following RZZ complex inhibition. The results shown here suggest that a direct function in chromosome orientation may contribute to the chromosome missegregation caused by the loss of kinetochore dynein following RZZ complex inhibition. We note that the Ndc80 module and chromokinesin, acting together in the absence of kinetochore dynein, can congress, biorient, and properly segregate a substantial

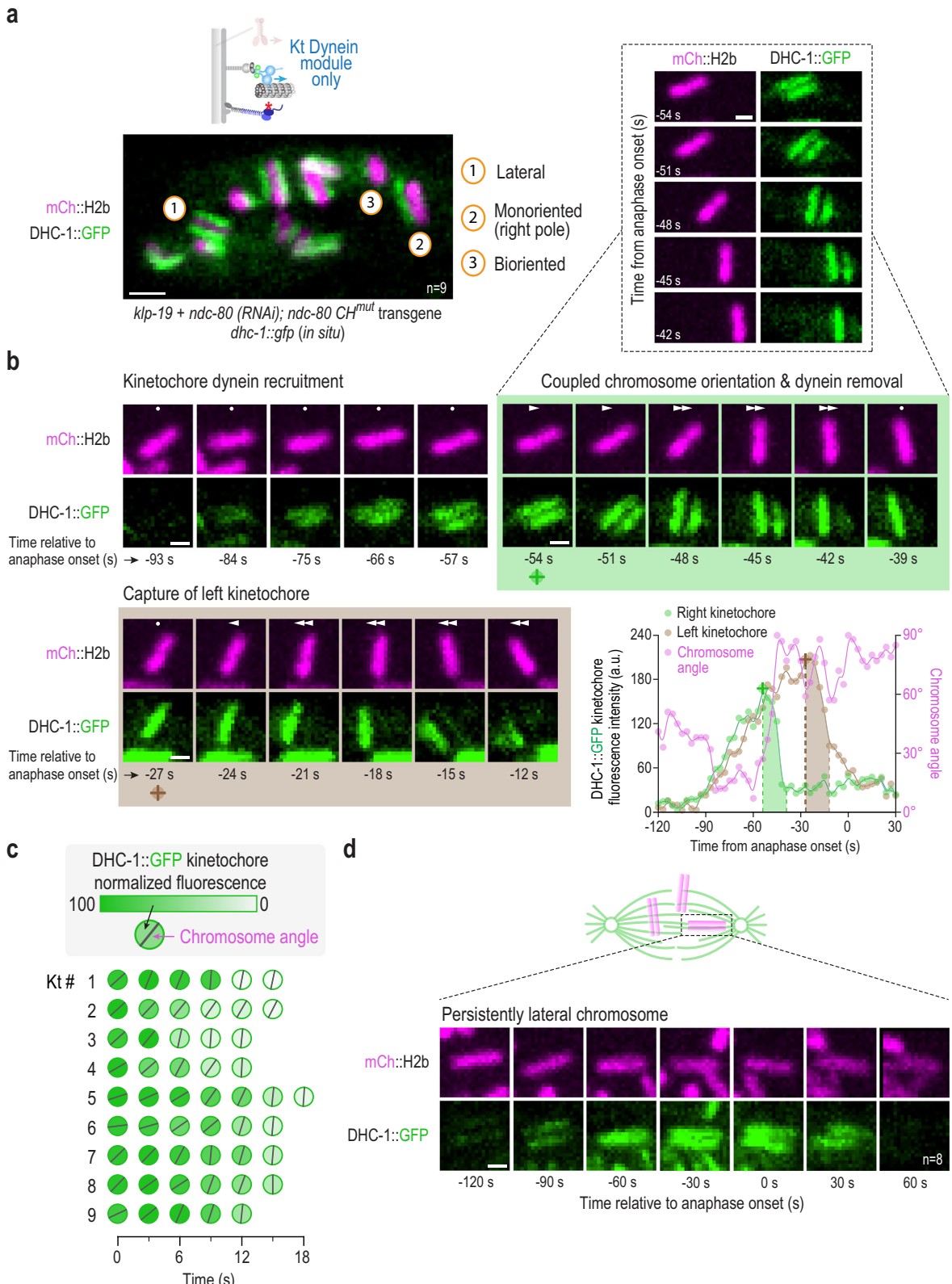

proportion of chromosomes (Supplementary Fig. 3b–d). Thus, with respect to the orientation function of chromosomal microtubule-targeted activities, the chromokinesin and Ndc80 module combination acts in parallel to the kinetochore dynein module (Fig. 6a), with the action of all three required to ensure that every chromosome is properly bioriented and accurately segregated. The dominant contribution of chromokinesin and the Ndc80 module likely accounts for

the relative mildness of segregation defects observed in the absence of kinetochore dynein. The comparative analysis additionally indicates that the chromokinesin and Ndc80 module combination drives chromosome congression, with no significant contribution from kinetochore dynein (Supplementary Fig. 3d). Parallel action of kinetochore dynein and the chromokinesin-Ndc80 module combination, with the latter playing a more significant role, likely explains why observing the

**Fig. 4 | Orientation-coupled removal of dynein from kinetochores in the kinetochore dynein module-only state. a** Snapshot of dynein localization, visualized using in situ-tagged DHC-1::GFP, at a time point when chromosomes are autonomously orienting on the spindle in the kinetochore dynein module-only state. Numbered circles highlight 3 chromosomes in different states. The genotype and RNAi conditions employed here and in (**b–d**) are shown below the image. Scale bar, 2 μm. *n* is the number of embryos analyzed. **b** Image sequence of a single chromosome and of dynein localized to its two sister kinetochores. To aid comparison, the panels are centered on the chromosome; dots (no movement) and arrows (directional movement) in each panel indicate chromosome movement relative to the spindle pole. Boxed region shows a subset of panels where the chromosome was not centered to highlight coupled poleward translocation and orientation associated with the capture of the right kinetochore. Green and brown shaded boxes highlight successive capture of the two sister kinetochores. Time is relative to anaphase onset in seconds. Graph on the lower right quantifies the DHC-1::GFP fluorescence intensity on each kinetochore, along with the chromosome angle relative to the spindle axis. Shaded areas on the graph correspond to the shaded boxes around image panel sets. Symbols below −54 s and −27 s panels serve as reference points linking image panels to the graph. Scale bars, 1 μm. **c** Analysis of kinetochore dynein fluorescence intensity and chromosome angle for 9 kinetochores. Shading in the circle indicates DHC-1::GFP fluorescence signal and the black line indicates the chromosome angle relative to the spindle axis. For simplicity, translocation on the spindle is not shown. **d** DHC-1::GFP localization on a chromosome that maintains a persistent lateral orientation through anaphase and does not biorient. Scale bar, 1 μm. *n* is the number of chromosomes analyzed. Source data are provided as a Source Data file.

ability of kinetochore dynein to efficiently orient chromosomes required establishment of the approach described here.

In vertebrates, the kinetochore dynein module is a central component of the fibrous corona, the outermost region of the kinetochore[23,48,49], and it has been implicated in microtubule-dependent removal of checkpoint proteins (along with itself) from the corona. A second dynein-independent step has been proposed to remove checkpoint complexes from the core kinetochore[50]. Polymerization of the RZZ complex, along with farnesylation of Spindly, is critical for corona assembly[23]. In the *C. elegans* embryo, the extent to which kinetochore dynein module localization is corona-like remains unclear: SPDL-1 lacks the farnesylation consensus site and the curved assemblies extending outward from the core outer kinetochore that are characteristic of fibrous coronas built on unattached kinetochores in vertebrates are not observed[48]. Nonetheless, in *C. elegans*, as well as *Drosophila* and vertebrates, loss of the kinetochore dynein module leads to chromosome missegregation[27,31,32,51,52], which we suggest is due to loss of the orientation function that was revealed when the other major microtubule-directed activities on mitotic chromosomes were absent.

A major question raised by the above observations is how the kinetochore dynein module is able to coordinately orient chromosomes and remodel the outer kinetochore. The dynein motor complex, together with its motility co-factor dynactin and activating adapters such as Spindly, exhibits minus end-directed motility along the microtubule lattice[53]. Such motility would lead to chromosomes moving poleward with kinetochore dynein laterally bound to the microtubule surface, as has been observed during the initial capture of spindle microtubules by kinetochores in vertebrate cells[54,55]. More generally, dynein at the kinetochore has long been proposed to aid lateral capture of microtubules[55]. However, lateral capture and minus end-directed motility in the kinetochore dynein-only state would cause chromosomes to cluster near the poles, rather than biorient throughout the spindle. We, therefore, suggest that an end-coupled state of the kinetochore dynein module may underlie both its orientation and remodeling activities (Fig. 6b). Biophysical analysis has shown that when a depolymerizing microtubule end reaches a lattice-bound dynein molecule, microtubule depolymerization is suppressed and the force from the depolymerizing end is transmitted to the cargo coupled to the motor[56,57]. Such an end-coupled dynein interaction could exert a torque on the chromosome cargo[58], leading to its rotation. Dynamically visualizing the geometry of kinetochore-microtubule interactions is necessary to explore this speculative idea. Unfortunately, due to the extremely fast dynamics (orientation occurs in 10–15 s) and the absence of discrete kinetochore fibers, this is currently technically unfeasible in the *C. elegans* embryo. One expectation from this line of thinking is that orientation will be kinetochore-autonomous, with a single kinetochore orienting the chromosome. The coupled analysis of chromosome and DHC-1 dynamics is consistent with this expectation. In addition, the contrast in DHC-1 behavior between neighboring laterally oriented and perpendicularly

bioriented chromosomes on the spindle lends support to the notion that an end-coupled dynein attachment state underlies both orientation and remodeling. As few prior studies have analyzed the interaction between the dynein motor and depolymerizing ends of dynamically unstable microtubules[56,57], in contrast to the myriad studies on dynein minus end-directed motility on artificially stabilized microtubules, more work is needed to understand how dynein interfaces with dynamic microtubule ends and whether mutations can be engineered that uncouple motility from the ability to suppress catastrophe and act as an end-coupler. Directly visualizing and manipulating the end-coupled kinetochore dynein state, which we propose plays a critical role during mitotic chromosome segregation, may also prove important in the context of other cellular dynein functions that involved its interactions with a dynamic microtubule network.

## Methods

### *C. elegans* strain maintenance

*C. elegans* strains are described in Supplementary Table S1. All strains were maintained using standard *C. elegans* nematode growth media (NGM) and seeded with *Escherichia coli* (OP50) at 20 °C to generate a feeding lawn. L4 stage *C. elegans* nematodes were passed for maintenance and plates were kept at 20 °C.

### CRISPR/Cas9-mediated genome editing

Endogenous tagging of the *klp-19* locus (Kinesin-Like Protein 19, chromokinesin and ortholog of human KIF4A), the *knl-1* locus (Kinetochore NuLl 1), the *dhc-1* locus (Dynein Heavy Chain), the *spdl-1* locus (SPinDLy (*Drosophila* chromosome segregation) homolog) and the *rod-1* locus (ROD (*Drosophila* RoughDeal) homolog) at the N- or C-terminus (see https://wormbase.org for more information) was performed using CRISPR/Cas9[59–62]. The specific method, guide RNA sequences, and homology arm sequences used to generate each strain are described in Supplementary Table S2. Briefly, a DNA mix or Cas9-RNP mix, containing the respective repair template, guide RNA sequences, Cas9, and selection markers were injected into young N2 adult nematodes. Recombinant strains were identified using the appropriate selection method and by genotyping PCR, and the sequences confirmed using Sanger sequencing of the edited genomic region.

### RNA-mediated interference

DNA templates were generated via PCR using the primers as specified in Supplementary Table S3. DNA templates were subsequently purified using a QIAquick PCR Purification Kit (Qiagen). Single-stranded RNA was generated from each DNA template using a MEGAscript™ T3 and T7 Transcription Kit (Invitrogen) and subsequently purified using a MEGAclear™ Transcription Clean-Up Kit (Invitrogen). Double-stranded RNA (dsRNA) was generated by annealing the single-stranded RNAs at 37 °C for 30 min[36]. 36–46 h before dissection and embryo imaging, the dsRNA was injected into L4 hermaphrodites, which were maintained at 20 °C. All RNAi experiments were performed

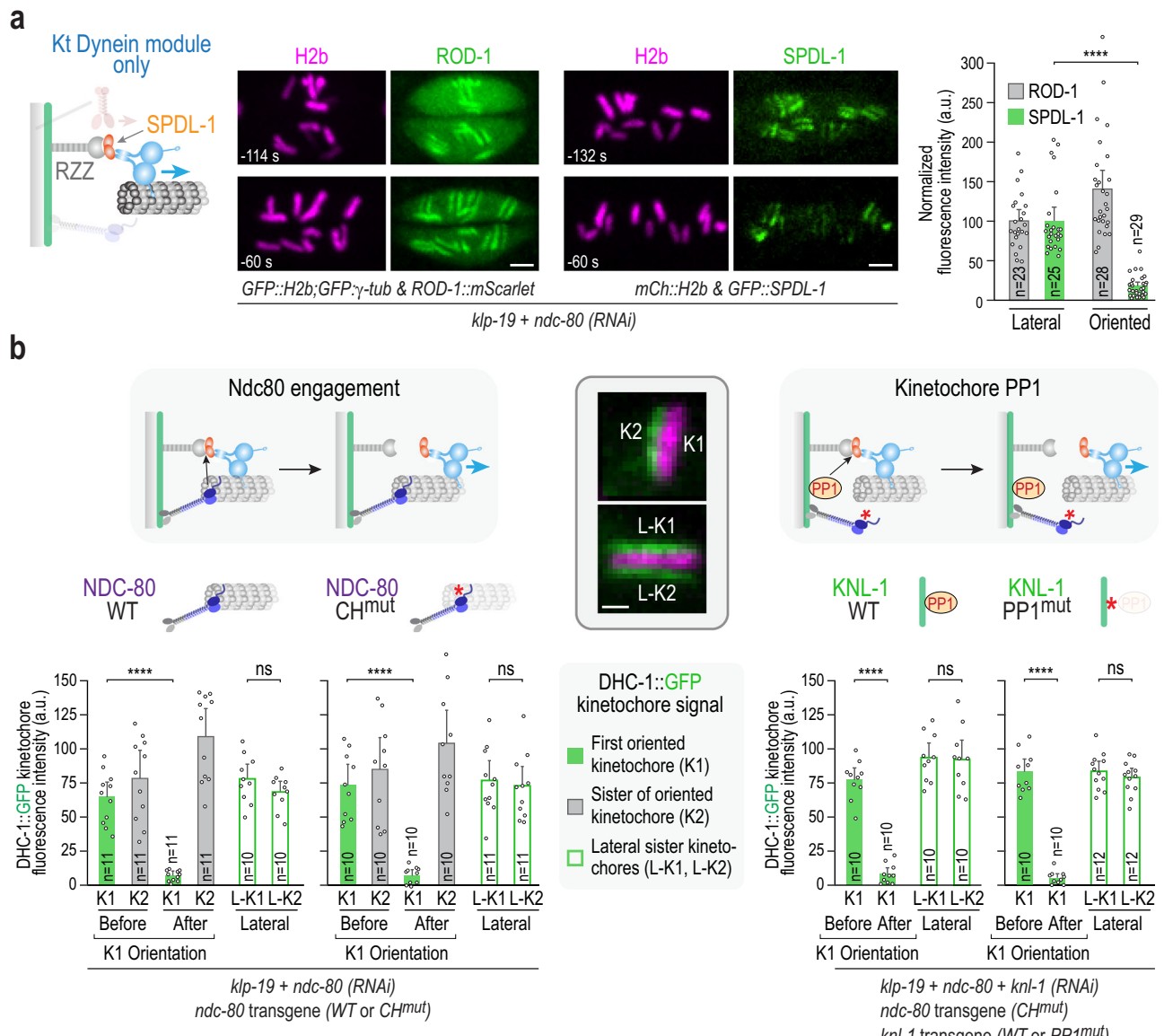

**Fig. 5 | Analysis of the trigger for dynein removal coupled to chromosome orientation in the kinetochore dynein module-only state. a** Localization of the RZZ complex and SPDL-1 in the kinetochore dynein module-only state. In situ-tagged ROD-1 and SPDL-1 were imaged; note that the transgene encoding CH^mut NDC80 was not present. Scale bars, 2.5 μm. Time listed is in seconds relative to anaphase onset. Graph on the right shows the quantification of ROD-1 and SPDL-1 signals on persistently lateral versus perpendicularly oriented chromosomes. Error bars are the 95% CIM. *n* represents the number of chromosomes analyzed. *p*-values are from two-tailed Student's *t*-tests for means: ****(*p* < 0.0001) & ns (not significant, *p* > 0.05) (SPDL-1; 6.13E-13). **b** Comparison of kinetochore dynein removal in the kinetochore dynein-only state with a functional or mutant Ndc80 module (left) or with or without kinetochore-localized protein phosphatase 1 (PP1; right). Schematics on top highlight models for the removal trigger; cartoons above the graphs indicate the perturbations analyzed in the kinetochore dynein-only state;

text below the graphs indicates the perturbations used to generate these states. DHC-1::GFP was quantified on sister kinetochores at the time point when the first sister was captured and began orienting toward a pole ("Before") and after full orientation was achieved ("After"); K1 and K2 refer to the first oriented kinetochore and its sister, respectively. For the KNL-1 WT versus PP1^mut comparison, DHC-1 signal was only measured after the orientation of the first sister was completed. DHC-1::GFP on sister kinetochores of chromosomes that maintained a persistently lateral orientation until anaphase onset was also measured (L-K1, L-K2). Central image insets show example images of K1, K2, and L-K1, L-K2; scale bar is 1 μm. Error bars are the 95% CIM. *n* is the number of sister kinetochore pairs (chromosomes) analyzed. *p*-values are from two-tailed Student's *t*-tests for means (NDC80 WT; 4.55E-9 and 0.18, NDC80 CH^mut; 1.95E-7 and 0.71, KNL-1 WT; 5.80E-11 and 0.89, KNL-1 PP1^mut; 6.76E-12 and 0.39). Source data are provided as a Source Data file.

using 1 mg/ml individual dsRNAs, or 1:1 or 1:1:1 mixtures of 1 mg/ml individual dsRNAs.

**Spinning-disk confocal fluorescence microscopy**
One-cell embryos were imaged on a spinning-disk confocal (Revolution XD Confocal System; Andor Technology) with a confocal scanner unit (CSU-10, Yokogawa Corporation) attached to an inverted microscope body (TE2000-E, Nikon), illuminated using solid-state 100 mW

lasers using either a 60× or 100 × 1.4 NA Plan Apochromat oil objective (Nikon) and an EMCCD camera (iXon DV887, Andor Technology, 16 × 16 μm pixel size) (Desai Lab, San Diego).

One-cell embryos were also imaged on a spinning-disk confocal (CSU-W1 Confocal System, Nikon) with a confocal scanner unit (CSU-W1, Yokogawa Corporation) attached to an inverted microscope body (ECLIPSE Ti2-E, Nikon), illuminated using solid-state 200 mW lasers using either a 60× or 100 × 1.4 NA HP Plan Apochromat oil objective

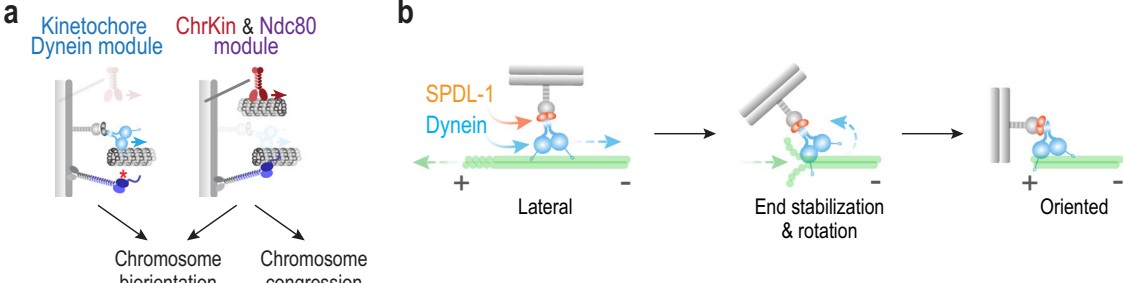

**Fig. 6 | The kinetochore dynein module acts in parallel to chromokinesin and the Ndc80 module to biorient chromosomes. a** Schematic summary highlighting the parallel action of the kinetochore dynein module and the combination of chromokinesin and the Ndc80 module in chromosome biorientation. Only the chromokinesin-Ndc80 module combination is able to drive congression to the spindle equator. **b** Speculative model, based on biophysical analysis of dynein engagement with depolymerizing microtubule ends, for how a dynamic microtubule end-coupled state may underlie the orientation function of kinetochore dynein. In brief, a rapidly depolymerizing spindle microtubule end reaches laterally bound kinetochore dynein, which suppresses depolymerization and transmits a force that rotates the chromosome to orient it toward a spindle pole.

(Nikon) and an sCMOS camera (Prime 95B, Teledyne Photometrics, 11 × 11 μm pixel size) (Cheerambathur Lab, Edinburgh).

### One-cell embryo sample preparation
One-cell embryos were dissected from adult hermaphrodites in M9 buffer, placed onto a microscope slide containing a 2% agarose in M9 pad, and subsequently covered with a 22 × 22 mm high-precision cover glass (No. 1.5H, Marienfeld).

### Imaging and localization analysis of in situ-tagged GFP fusions
For localization analysis of NDC80::GFP, GFP::KLP-19, and DHC-1::GFP, 5 × 1.5 μm z-stacks were acquired every 10 s, and for KNL-1::GFP and GFP::MAD-1 every 3 s. For localization analysis of GFP::SPDL-1 and ROD-1::mScarlet-I, 7 × 1.0 μm z-stacks were acquired every 3 s or 6 s. Maximum intensity projections were generated using ImageJ2 (Fiji) and subsequently, the fluorescent background was subtracted.

### Chromosome dynamics and minimum bounding box analysis
For minimum bounding box (MBB) analysis, 5 × 1.5 μm z-stacks were acquired every 3 s and maximum intensity projections were generated using ImageJ2 (Fiji) and rotated to position the spindle poles horizontally. Fluorescence intensity for all maximum intensity projections in the series was normalized, converted to 8-bit and the fluorescent background subtracted. Subsequently, to each maximum intensity projected image, an MBB was fitted (pixel value > 0, i.e. fluorescent signal from GFP::H2b) to measure chromosome positioning with width (and height) of the MBB determined by the positioning of the outermost chromosomes.

### Chromosome orientation analysis
For chromosome orientation analysis, 5 × 1.5 μm or 7 × 1.0 μm z-stacks were acquired every 3 s for GFP::H2b, maximum intensity projections generated using ImageJ2 (Fiji) and rotated to position the spindle poles horizontally. Subsequently, chromosome angles were determined by fitting a line along each chromosome axis and measuring the smallest angle between the chromosome axis and spindle pole-to-pole axis. Angles were read out for all chromosomes visible in the maximum intensity projection varying from 0 degrees (chromosome parallel to pole-to-pole axis, i.e., −) to 90 degrees (chromosome perpendicular to pole-to-pole axis, i.e., |). The data was binned using 15-degree bins, and data from -15 s pre NEBD and anaphase onset pooled and plotted as a percentage of chromosomes. The positions of these chromosomes were not measured except for Fig. 1d, where the x-axis represents the spindle pole-to-pole axis (with the origin set to the spindle equator) and the centers of the chromosomes projected onto this axis. This data was binned using 30-degree bins and plotted.

### Chromosome biorientation analysis
For chromosome biorientation analysis, 5 × 1.5 μm z-stacks were acquired every 10 s to image KNL-1::GFP and mCherry::HIS-58. Next, maximum intensity projections were generated using ImageJ2 (Fiji) and the spindle rotated based on its diffuse autofluorescent signal to position it horizontally. A chromosome was scored as 'biorientated' when KNL::GFP signal was visible on either side of the mCherry::HIS-58 signal with each individual kinetochore facing a spindle pole at 20 s before anaphase onset.

### Single-chromosome dynamics and localization analysis
For single-chromosome localization analysis of DHC-1::GFP, 7 × 1.0 μm z-stacks were acquired every 3 s starting 1 min after NEBD, and maximum intensity projections were generated using ImageJ2 (Fiji). Subsequently, a rectangular box (0.4 × 1.6 μm) fitted adjacent to the mCherry::H2B signal (chromosome) encapsulating kinetochore dynein to obtain the average DHC-1::GFP intensity and the fluorescent background subtracted. For the data of Fig. 4b, c chromosome angles were determined by fitting a line along each chromosome axis and measuring the smallest angle between the chromosome axis and spindle pole-to-pole axis, varying from 0 degrees (chromosome parallel to pole-to-pole axis, i.e., −) to 90 degrees (chromosome perpendicular to pole-to-pole axis, i.e., |). Qualitative movement (as depicted by the dots and arrows) was determined based on the chromosome position in the frame preceding and the one succeeding. A "capture" (also called "before") was defined as the time point preceding consistent chromosome poleward movement and its coinciding orientation, and "after" when the chromosome was static and full orientation was achieved.

### Single-chromosome kinetochore-dynein distribution analysis
For analysis of the kinetochore-dynein distribution of DHC-1::GFP, 7 × 1.0 μm z-stacks were acquired every 3 s starting 1 min after NEBD, and maximum intensity projections were generated using ImageJ2 (Fiji). Subsequently, a 3-pixel-wide line (0.55 μm × 2.0 μm) was fitted adjacent and parallel to the mCherry::H2b signal (chromosome), with position 0.0 μm corresponding to the bottom of the chromosome and position 2.0 μm to the top, to obtain the line intensity profiles for DHC-1::GFP. Subsequently, the fluorescent background was subtracted and the intensities of the 3 pixels covering the line width averaged.

### Quantification and statistical analysis
ImageJ2 (Fiji) (version 2.14.0/1.54 f) was used to extract quantitative information from image series, and processed using Excel, GraphPad Prism, and/or OriginPro 2022b ((64-bit) SR1 9.9.5.171). Measurements were taken from distinct samples and assumed to be normally

distributed. Statistical analysis was done using OriginPro 2022b ((64-bit) SR1 9.9.5.171) by performing two-tailed Student's $t$-tests for means with ****($p < 0.0001$) & ns (not significant, $p > 0.05$). No statistical method was used to predetermine the sample size. No data were excluded from the analyses. The experiments were not randomized. The Investigators were not blinded to allocation during experiments and outcome assessment.

## Reporting summary

Further information on research design is available in the Nature Portfolio Reporting Summary linked to this article.

## Data availability

Data supporting this study are available in the primary manuscript and the supplemental material files. The *C. elegans* constructs and plasmids and the raw imaging data are available upon request from the lead authors. All data plotted in the graphs are provided in the Source Data file. Source data are provided with this paper.

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

## Acknowledgements
We thank Reto Gassmann and François Nédélec for helpful discussions and Sander van den Heuvel for sharing strains. This work is supported by an NIH grant to A.D (R01 GM074215), a Sir Henry Wellcome Postdoctoral Fellowship (215925) to B.P., a Wellcome Sir Henry Dale Fellowship (208833) to D.K.C. and a Wellcome Principal Research Fellowship (107022) to W.C.E. Part of the imaging was performed in Centre Optical Instrumentation Laboratory (COIL), which is supported by a Core Grant (203149) to the Wellcome Centre for Cell Biology at the University of Edinburgh. Currently, B.P. is sponsored by W.C.E. A.D. acknowledges salary support from the Ludwig Institute for Cancer Research.

## Author contributions
B.P. and A.D. conceived the study. B.P., D.K.C., and A.D. designed the experiments. B.P. conducted and analyzed the experiments. B.P. and D.K.C. generated the *C. elegans* strains. B.P., W.C.E., D.K.C., and A.D. prepared the manuscript.

## Competing interests
The authors declare no competing interests.
