## [Peer Review File · Nature Communications]

REVIEWER COMMENTS

Reviewer #1 (Remarks to the Author):

By co-depleting KNL-1 (essential for outer kinetochore assembly in *C. elegans*) and the KIF4 family chromokinesin KLP-19 (the major mitotic chromatin-localized motor) the authors generated a blank slate with respect to microtubule interactions on mitotic chromosomes. They then reconstituted three fundamental branches of kinetochore (KT)-microtubule (MT) attachment by selectively re-introducing specific functions. Specifically, they created a chromokinesin-only state by depleting KNL-1 (“ChrKin only”); an NDC-80-only state (“Ndc80 module only”) by co-depleting KLP-19 and ROD-1 (ROD-1 is a subunit of the RZZ complex that recruits dynein to kinetochores); and a kinetochore dynein module-only (“Kt Dynein module 80 only”) state by co-depleting KLP-19 and NDC-80; in this case, the authors also expressed a mutant form of NDC-80 that disrupts microtubule binding but preserves other functions of the NDC-80 complex). The authors use two main criteria for evaluating chromosome behavior upon introduction of these perturbations, namely chromosome dispersion along the spindle axis and their orientation relative to the same. The main (and surprising) conclusion of the work is that the Dynein module, albeit insufficient for chromosome congression, is sufficient to promote chromosome bi-orientation and to remodel the outer kinetochore to turn off checkpoint signaling. Conversely, a combination of the ChrKin and Ndc80 modules promotes congression and apparent bi-orientation, but the frequency of mis-segregation events at anaphase skyrockets. Remarkably, the Ndc80 complex is on its own largely unable to promote biorientation, and only achieving a pseudo-congression with frequent chromosome segregation failure at anaphase.

Overall, the reconstitution experiments described here support several unanticipated conclusions on the role of kinetochore dynein. The manuscript will be of great interest to the community working on this important cellular machine. The experiments are extremely well controlled. For instance, the role of the Dynein module is validated with two controls conditions that selectively eliminate Dynein from kinetochores (expression of a Spindly mutant or depletion of ROD). The Figures are largely self-explanatory and greatly facilitate reading the manuscript.

In conclusion, I strongly support publication of the manuscript. Nonetheless, I would like to ask the authors to consider the following points.

Major point

In Figure 3D, it is hard to recognize any real “anaphase” chromosome behavior. The distance between sisters does not seem to change at all in subsequent frames. Is this indicative of extensive

merotelic attachment or mitotic arrest, as suggested by the persistence of ROD (Figure 4. Incidentally, would ROD rather leave the KT before anaphase if the Ndc80 module were present)? What exactly defines “anaphase” in this system? Is it spindle elongation?

Minor points:

Lines 14-18: This sentence is very long, and I would recommend splitting it or at least adding “and” between “chromosomes” and “that direct” on line 14.

Line 31: I recommend splitting the sentence after “CENP-T” and completing the subsequent sentence with “is also present”

Line 284: “in vivo”, not “in vitro”

Reviewer #2 (Remarks to the Author):

In this study from Prevo et al. the authors dissect the individual contributions of different microtubule-associated protein complexes on chromosome orientation and segregation during mitosis. Using RNAi-based depletions and microscopy-based analyses in *C. elegans* early embryos, they directly or indirectly deplete the major force generators from kinetochores to create what they refer to as a “blank slate” that completely lacks microtubule-binding activities. By comparing states with only one force generator present (chromokinesin-, Ndc80 module-, or dynein module-only), they assess their individual effects on chromosome congregation, orientation, and segregation. The authors find that only the dynein module-only state exhibits chromosomes that are oriented perpendicularly to the spindle axis and are bioriented. This behavior is further dependent on the ability of SPDL-1 to recruit dynein, suggesting that dynein specifically is needed for chromosome biorientation in this system. Finally, the authors examine dynein module protein dynamics and observe kinetochore-autonomous and orientation-coordinated localization of dynein and SPDL1, but not ROD-1. The observed loss of DHC-1 from oriented chromosomes was independent of Ndc80 and PP1. The authors speculate a model in which MT depolymerization leads to dynein and chromosome rotation.

Overall, the key findings in this paper are that dynein is sufficient for the biorientation of chromosomes in *C. elegans* and that SPDL-1 and DHC-1 removal from kinetochores occurs in an orientation-coordinated manner. Although some mechanistic details of this role for dynein remain to be elucidated, the experiments in this paper are rigorous and offer novel insights into the role of dynein in chromosome segregation. Several suggestions for consideration are provided below, particularly pertaining to the presentation of the work and some of the conclusions drawn.

Major points:

1. We recommend that the authors use caution when generalizing the distinct “phases” in which each of the different components act. It’s likely that these components rely (directly and/or indirectly) on the action of other proteins that are missing in these states in order to properly function at the right time. Thus, the dynamics are most likely impaired in each of the states in different ways. Therefore, it seems inaccurate to 1. conclude that these phases indicate the time of action in normal conditions or 2. map the phases relative to each other, as shown in the final model. It is informative to mention the phases, but that more caution should be exercised when generalizing these observations.

2. The authors comment that “We note that the Ndc80 module and chromokinesin, acting together in the absence of kinetochore dynein, can congress, biorient and properly segregate a substantial proportion of chromosomes (Fig. S2B,C). Thus, with respect to the orientation function of chromosomal microtubule-targeted activities, the chromokinesin and Ndc80 module combination acts in parallel to the kinetochore dynein module, with the action of all three required to ensure that every chromosome is properly bioriented and accurately segregated”. Although dynein is able to orient chromosomes in the dynein module only-state, that does not necessarily indicate that it always does so under normal conditions in parallel with the Ndc80 module and chromokinesin. It may be that this serves more of a regulatory function or becomes more important under certain conditions. Perhaps it even becomes the dominant module in some circumstances. Given that this newly discovered ability for dynein is a main finding in this paper, we suggest discussing the potential physiological importance more in depth, but also maintaining a degree of uncertainty since it hasn’t been directly tested across diverse conditions.

3. The authors describe their functional dissection of the different microtubule binding activities as “an *in vivo* reconstitution approach”. This terminology implies that they have eliminated things completely and then are selectively adding them back, for example through the ectopic targeting of specific proteins to kinetochores. However, this is not what is occurring here. Instead, the authors are comparing several selected and well-designed depletion conditions to compare different kinetochore states. This “reconstitution” terminology is unnecessarily sold. As this doesn’t accurately describe their experiments, the authors should reword this throughout.

4. The observed behaviors for the roles of the different microtubule binding proteins and force generators are rigorous and interesting. However, it remains possible that some of these behaviors reflect functional properties that are specific to holocentric chromosomes and or the nature of the *C. elegans* kinetochore, which lacks some microtubule binding activities present in other eukaryotes. The nature of the organism and system isn't mentioned at all in the title or abstract, which may be misleading to some superficial readers. It would be helpful if the authors were to clarify this point and mention this potential caveat throughout the text, including a reference in the abstract.

Minor points:

1. Since the paper does not include a standalone introduction, it would be helpful to include some additional background information and also move some of the background information that is already provided to earlier in the text. Specifically, it would be beneficial to include information regarding what is already known about the specific roles/mechanisms of each of the components being investigated in the functional depletion assay before the results are presented.

2. Multiple sentences would benefit from restructuring to improve clarity and readability, particularly for a general audience not familiar with these proteins or field (for example, see sentences starting at lines 2, 23, and 74).

3. In general, most figures clearly indicate the *C. elegans* embryo genotype for each experiment with text labels and often helpful diagrams. However, there are a couple instances (Fig 4B, C, D) where the genotype is not specified and needs to be denoted in either the figure panel or in the figure legend.

4. When discussing the phases in figure 2, it would be helpful to explain how the phase is determined. This is described in the methods section, but would be helpful to also include their logic in the body of the text.

Reviewer #3 (Remarks to the Author):

This manuscript by Prevo et al. dissects three conserved motile activities that align and congress chromosomes to the metaphase plate in the roundworm *C. elegans*. The activities in question are the chromokinesin Klp-19 that pushes chromosomes toward the spindle equator, kinetochore-localized dynein that is expected to pull chromosomes towards the spindle poles, and the Ndc80 complex, which couples chromosome movement to microtubule depolymerization. To observe the individual activities, the authors use a well-designed approach that knocks-down or inactivates each activity either individually or in combination. Their analysis clarifies the roles and coordination of the three activities to chromosome alignment and congression. The authors also document a surprisingly potent role for kinetochore-localized dynein and propose that dynein's ability to "capture" depolymerizing plus-ends, rather than its minus-end directed motility, is involved in this process. Finally, the authors show that the release of the kinetochore-localized dynein does not require Ndc80- or Knl1-mediated regulation.

The study is well-designed and executed, and it clarifies the processes that chromosome alignment and congression specific to *C. elegans*. The finding regarding dynein's release from the kinetochore is an important one and broadly applicable to other metazoan systems. The proposal that the kinetochore-localized dynein engages microtubule plus-ends is novel and important. Therefore, I support the publication of this manuscript. My main criticism is that the authors don't fully test the mechanism that detects chromosome alignment and signals for the release of the kinetochore-localized dynein. I suggest, for the authors to consider, a couple of experiments that may greatly increase the impact of this work.

1. The release of dynein from the kinetochore likely involves phosphoregulation because of the way it is released in anaphase, as the authors note. The authors show that Knl1-mediated PP1 recruitment is not involved in this release. Given the authors' proposal that dynein's plus-end capturing activity is responsible for rotating and aligning chromosomes, the regulatory signal for dynein's removal may also be delivered from the microtubules by a motor, specifically Kif18. The kinesin Kif18 contains a PP1 recruitment site (Wever et al. *Biophysical and Biochemical research communications* 2014; also in fission yeast – Meadows et al. *Dev Cell* from the Millar lab), enriches at the plus-ends of k-fibers (Stumpff et al. *Dev Cell*), and contributes to SAC silencing (Janssen et al. *Current Biology* 2018, also in fission yeast. There are likely to be other, more appropriate references; I include here what a shallow search revealed). Assuming that PP1-recruitment via mitotic kinesins is conserved, this model could be tested using the appropriate mutants.

2. To my eye, in video 5 and 6 the chromosomes with the dynein-only module show a striking "bounce-back" when the first kinetochore becomes perpendicular to the spindle axis and loses its dynein, and, maybe, again after the second one loses its dynein. I am wondering if this hunch is borne out by careful quantitation. If true, this opens up another possibility: the signal for the

removal of kinetochore-only dynein may be the ability of the kinetochore-bound dynein to produce force and processively walk towards the minus end.

3. Is it feasible to observe chromosome rotation prior to dynein-mediated alignment? Such an observation will also further support the model of chromosome alignment by dynein-plus-end interaction.

Minor comments:

1. For the wider audience, it will be useful to include a cartoon cross-section of the chromosome to further clarify the localization of the three modules in Figure 1A.

2. Figure 1E cartoon: Conventionally, angles are measured counterclockwise. Because the spindle axis is the reference axis, it would be more appropriate to tilt the chromosome the other way and then indicate the angle measurement.

3. The percentage scale on many of the chromosome orientation plots changes (e.g., 100% and 80% in Figure 1E). It would ideal to maintain the same scale even if the bars become small.

4. The micrographs and videos also give the impression that the kinetochore-bound dynein is released locally from the side that appears to align first. It would be useful to document this observation because it further supports the notion that the dynein removal cue is dependent on a specific form of dynein-microtubule interaction.

Reviewer #1 (Remarks to the Author):

By co-depleting KNL-1 (essential for outer kinetochore assembly in *C. elegans*) and the KIF4 family chromokinesin KLP-19 (the major mitotic chromatin-localized motor) the authors generated a blank slate with respect to microtubule interactions on mitotic chromosomes. They then reconstituted three fundamental branches of kinetochore (KT)-microtubule (MT) attachment by selectively re-introducing specific functions. Specifically, they created a chromokinesin-only state by depleting KNL-1 (“ChrKin only”); an NDC-80-only state (“Ndc80 module only”) by co-depleting KLP-19 and ROD-1 (ROD-1 is a subunit of the RZZ complex that recruits dynein to kinetochores); and a kinetochore dynein module-only (“Kt Dynein module 80 only”) state by co-depleting KLP-19 and NDC-80; in this case, the authors also expressed a mutant form of NDC-80 that disrupts microtubule binding but preserves other functions of the NDC-80 complex). The authors use two main criteria for evaluating chromosome behavior upon introduction of these perturbations, namely chromosome dispersion along the spindle axis and their orientation relative to the same. The main (and surprising) conclusion of the work is that the Dynein module, albeit insufficient for chromosome congression, is sufficient to promote chromosome bi-orientation and to remodel the outer kinetochore to turn off checkpoint signaling. Conversely, a combination of the ChrKin and Ndc80 modules promotes congression and apparent bi-orientation, but the frequency of mis-segregation events at anaphase skyrockets. Remarkably, the Ndc80 complex is on its own largely unable to promote biorientation, and only achieving a pseudo-congression with frequent chromosome segregation failure at anaphase.

Overall, the reconstitution experiments described here support several unanticipated conclusions on the role of kinetochore dynein. The manuscript will be of great interest to the community working on this important cellular machine. The experiments are extremely well controlled. For instance, the role of the Dynein module is validated with two controls conditions that selectively eliminate Dynein from kinetochores (expression of a Spindly mutant or depletion of ROD). The Figures are largely self-explanatory and greatly facilitate reading the manuscript.

In conclusion, I strongly support publication of the manuscript. Nonetheless, I would like to ask the authors to consider the following points.

We thank the reviewer for their positive assessment of the work.

Major point

In Figure 3D, it is hard to recognize any real “anaphase” chromosome behavior. The distance between sisters does not seem to change at all in subsequent frames. Is this indicative of extensive merotelic attachment or mitotic arrest, as suggested by the persistence of ROD (Figure 4. Incidentally, would ROD rather leave the KT before anaphase if the Ndc80 module were present)? What exactly defines “anaphase” in this system? Is it spindle elongation?

To respond to this feedback, we clarify that our only goal was to highlight that the movement of sisters apart was indicative of chromosomes having achieved a bioriented state. Anaphase chromosome separation in the *C. elegans* embryo is almost exclusively driven by spindle elongation (PMID: 11402065). In the “Kinetochore Dynein-Only” state, there is premature spindle elongation as the Ndc80 module is non-functional, and hence normal anaphase separation is not expected. Based on the reviewer’s comment, we added a GFP::H2b chromosome time-series, which better highlights sister separation (**new panel added to Fig.**

3d). We have also adjusted the text to avoid implying that normal anaphase-like chromosome behavior is observed.

Minor points:

Lines 14-18: This sentence is very long, and I would recommend splitting it or at least adding “and” between “chromosomes” and “that direct” on line 14.

We have split this sentence as recommended.

Line 31: I recommend splitting the sentence after “CENP-T” and completing the subsequent sentence with “is also present”

We have edited the sentence in question as recommended.

Line 284: “in vivo”, not “in vitro”

Thank you – this has been corrected.

Reviewer #2 (Remarks to the Author):

In this study from Prevo et al. the authors dissect the individual contributions of different microtubule-associated protein complexes on chromosome orientation and segregation during mitosis. Using RNAi-based depletions and microscopy-based analyses in *C. elegans* early embryos, they directly or indirectly deplete the major force generators from kinetochores to create what they refer to as a “blank slate” that completely lacks microtubule-binding activities. By comparing states with only one force generator present (chromokinesin-, Ndc80 module-, or dynein module-only), they assess their individual effects on chromosome congregation, orientation, and segregation. The authors find that only the dynein module-only state exhibits chromosomes that are oriented perpendicularly to the spindle axis and are bioriented. This behavior is further dependent on the ability of SPDL-1 to recruit dynein, suggesting that dynein specifically is needed for chromosome biorientation in this system. Finally, the authors examine dynein module protein dynamics and observe kinetochore-autonomous and orientation-coordinated localization of dynein and SPDL1, but not ROD-1. The observed loss of DHC-1 from oriented chromosomes was independent of Ndc80 and PP1. The authors speculate a model in which MT depolymerization leads to dynein and chromosome rotation.

Overall, the key findings in this paper are that dynein is sufficient for the biorientation of chromosomes in *C. elegans* and that SPDL-1 and DHC-1 removal from kinetochores occurs in an orientation-coordinated manner. Although some mechanistic details of this role for dynein remain to be elucidated, the experiments in this paper are rigorous and offer novel insights into the role of dynein in chromosome segregation. Several suggestions for consideration are provided below, particularly pertaining to the presentation of the work and some of the conclusions drawn.

We thank the reviewer for their interest in the work and for suggestions on improving the presentation and clarifying the conclusions.

Major points:

1. We recommend that the authors use caution when generalizing the distinct “phases” in which each of the different components act. It’s likely that these components rely (directly and/or indirectly) on the action of other proteins that are missing in these states in order to properly function at the right time. Thus, the dynamics are most likely impaired in each of the states in different ways. Therefore, it seems inaccurate to 1. conclude that these phases indicate the time of action in normal conditions or 2. map the phases relative to each other, as shown in the final model. It is informative to mention the phases, but that more caution should be exercised when generalizing these observations.

Thank you for this valuable feedback. We agree with the concerns articulated and have modified the figures and text to avoid implying that components act only in specific phases. First, we removed the schematic with the phases in the final model (as our most significant conclusion relates to the role of the kinetochore dynein module in chromosome orientation, we focused on that aspect). Second, we included imaging of the three components (all tagged at their endogenous locus with GFP) in *Fig. S1* to highlight their temporal localization dynamics at kinetochores. Third, we removed the shading highlighting “phases” in *Fig. 2b*. Finally, we have removed all text implying that the analyzed components act in specific phases throughout the manuscript (as well as in the Methods).

2. The authors comment that “We note that the Ndc80 module and chromokinesin, acting together in the absence of kinetochore dynein, can congress, biorient and properly segregate a substantial proportion of chromosomes (Fig. S2B,C). Thus, with respect to the orientation function of chromosomal microtubule-targeted activities, the chromokinesin and Ndc80 module combination acts in parallel to the kinetochore dynein module, with the action of all three required to ensure that every chromosome is properly bioriented and accurately segregated”. Although dynein is able to orient chromosomes in the dynein module only-state, that does not necessarily indicate that it always does so under normal conditions in parallel with the Ndc80 module and chromokinesin. It may be that this serves more of a regulatory function or becomes more important under certain conditions. Perhaps it even becomes the dominant module in some circumstances. Given that this newly discovered ability for dynein is a main finding in this paper, we suggest discussing the potential physiological importance more in depth, but also maintaining a degree of uncertainty since it hasn’t been directly tested across diverse conditions.

Based on this feedback, in the restructured manuscript we include a section in the Discussion focused on the significance of the orientation function of kinetochore dynein revealed by our analysis. We note that across a wide evolutionary spectrum in metazoans, removal of kinetochore dynein (e.g. through Rod mutation/depletion) is always associated with chromosome orientation and segregation defects (manifested as lagging chromosomes in anaphase). We know from analysis in *Drosophila* (PMID: 17417628) & *C. elegans* (e.g. **Fig. S3e,f**) that this missegregation is not explained by the involvement of the RZZ complex in spindle checkpoint signaling. Our results, which demonstrate the ability of the kinetochore dynein module to orient chromosomes on its own, lead us to suggest that kinetochore dynein may always contribute to chromosome orientation, even when the Ndc80 module and chromokinesin are active. While the Ndc80 module-chromokinesin combination is able to orient a significant number of the chromosomes, the kinetochore dynein module is required to ensure that all chromosomes biorient. We acknowledge that assessing this will require directed experimentation in other systems, potentially in *Drosophila* where inactivation of the spindle checkpoint (e.g. through genetic loss of Mad2) does not cause significant chromosome missegregation.

3. The authors describe their functional dissection of the different microtubule binding activities as “an in vivo reconstitution approach”. This terminology implies that they have eliminated things completely and then are selectively adding them back, for example through the ectopic targeting of specific proteins to kinetochores. However, this is not what is occurring here. Instead, the authors are comparing several selected and well-designed depletion conditions to compare different kinetochore states. This “reconstitution” terminology is unnecessarily used. As this doesn’t accurately describe their experiments, the authors should reword this throughout.

We have significantly edited the text to reduce mention of the approach and focus instead on the key findings. We do however find that use of this descriptor has broad appeal when presented and enhances the impact of the work by distinguishing it from single perturbation experiments.

4. The observed behaviors for the roles of the different microtubule binding proteins and force generators are rigorous and interesting. However, it remains possible that some of these behaviors reflect functional properties that are specific to holocentric chromosomes and or the nature of the *C. elegans* kinetochore, which lacks some microtubule binding activities present in other eukaryotes. The nature of the organism and system isn’t mentioned at all in the title or abstract, which may be misleading to some superficial readers. It would be helpful if the authors were to clarify this point and mention this potential caveat throughout the text, including a reference in the abstract.

The revised manuscript has been significantly reformatted to include an Introduction and describe the experimental model. We have also added titles for Results sections, which highlight that the model being used for the analysis is the *C. elegans* embryo. We, however, do not agree that working with *C. elegans* represents a “potential caveat”. The *C. elegans* embryo is a remarkable machine that biorients and segregates 12 mitotic chromosomes in 3 minutes and, in less than a day, supports the cell divisions necessary to build an entire organism with all of the major tissue types. In the Introduction of the revised manuscript, we note the holocentric nature of *C. elegans* chromosomes (also depicted in the schematic of *Fig. 1a*) and emphasize that all of the components present at *C. elegans* kinetochores that direct chromosome biorientation and segregation are conserved through humans. In addition, we highlight that the simpler composition of the *C. elegans* kinetochore (which is not specific to its holocentric nature; e.g. *Drosophila* has similarly simplified kinetochore composition) facilitates the type of experiments that are presented here. In terms of the title and abstract, we do not have space to include the full species name but the changes made to the revised manuscript indicate throughout that the work is being performed using the *C. elegans* embryo as a model.

Minor points:

1. Since the paper does not include a standalone introduction, it would be helpful to include some additional background information and also move some of the background information that is already provided to earlier in the text. Specifically, it would be beneficial to include information regarding what is already known about the specific roles/mechanisms of each of the components being investigated in the functional depletion assay before the results are presented.

The revised manuscript now includes an Introduction in which current thinking on the roles of components that are the focus of the analysis is summarized.

2. Multiple sentences would benefit from restructuring to improve clarity and readability, particularly for a general audience not familiar with these proteins or field (for example, see sentences starting at lines 2, 23, and 74).

We have revised the sentences highlighted as well as numerous other sections of the text to improve clarity and readability.

3. In general, most figures clearly indicate the *C. elegans* embryo genotype for each experiment with text labels and often helpful diagrams. However, there are a couple instances (Fig 4B, C, D) where the genotype is not specified and needs to be denoted in either the figure panel or in the figure legend.

We have revised the legend of **Fig. 4** to indicate that the genotype and RNAi conditions shown in panel **4a** apply to all of the other panels in the figure. We have also checked other figures and legends to confirm that the genotypes and experimental conditions are clearly indicated.

4. When discussing the phases in figure 2, it would be helpful to explain how the phase is determined. This is described in the methods section, but would be helpful to also include their logic in the body of the text.

As noted above, based on the reviewer's feedback we have removed the mention of "phases" from the revised manuscript. This has helped focus the text on the major findings related to the kinetochore dynein module.

Reviewer #3 (Remarks to the Author):

This manuscript by Prevo et al. dissects three conserved motile activities that align and congress chromosomes to the metaphase plate in the roundworm *C. elegans*. The activities in question are the chromokinesin Klp-19 that pushes chromosomes toward the spindle equator, kinetochore-localized dynein that is expected to pull chromosomes towards the spindle poles, and the Ndc80 complex, which couples chromosome movement to microtubule depolymerization. To observe the individual activities, the authors use a well-designed approach that knocks-down or inactivates each activity either individually or in combination. Their analysis clarifies the roles and coordination of the three activities to chromosome alignment and congression. The authors also document a surprisingly potent role for kinetochore-localized dynein and propose that dynein's ability to "capture" depolymerizing plus-ends, rather than its minus-end directed motility, is involved in this process. Finally, the authors show that the release of the kinetochore-localized dynein does not require Ndc80- or Knl1-mediated regulation.

The study is well-designed and executed, and it clarifies the processes that chromosome alignment and congression specific to *C. elegans*. The finding regarding dynein's release from the kinetochore is an important one and broadly applicable to other metazoan systems. The proposal that the kinetochore-localized dynein engages microtubule plus-ends is novel and important. Therefore, I support the publication of this manuscript. My main criticism is that the authors don't fully test the mechanism that detects chromosome alignment and signals for the release of the kinetochore-localized dynein. I suggest, for the authors to consider, a couple of experiments that may greatly increase the impact of this work.

We thank the reviewer for the positive assessment of the work and the recommendations to improve its impact.

1. The release of dynein from the kinetochore likely involves phosphoregulation because of the way it is released in anaphase, as the authors note. The authors show that Knl1-mediated PP1 recruitment is not involved in this release. Given the authors' proposal that dynein's plus-end capturing activity is responsible for rotating and aligning chromosomes, the regulatory signal for dynein's removal may also be delivered from the microtubules by a motor, specifically Kif18. The kinesin Kif18 contains a PP1 recruitment site (Wever et al. Biophysical and Biochemical research communications 2014; also in fission yeast – Meadows et al. Dev Cell from the Millar lab), enriches at the plus-ends of k-fibers (Stumpff et al. Dev Cell), and contributes to SAC silencing (Janssen et al. Current Biology 2018, also in fission yeast. There are likely to be other, more appropriate references; I include here what a shallow search revealed). Assuming that PP1-recruitment via mitotic kinesins is conserved, this model could be tested using the appropriate mutants.

We have investigated in-depth the kinetochore recruitment of protein phosphatases in the early *C. elegans* embryo (PP1; PP2A & PP4; PMIDs: 28698300, 36719399). To date, the only phosphatase activity that is robustly recruited to kinetochores is PP1 and the dominant recruitment site is in the KNL-1 N-terminus. Unlike vertebrates, *C. elegans* does not have a Kif18 family kinesin with a PP1 recruitment site. More importantly, using imaging of endogenous locus-tagged PP1, mutation of the PP1-binding RVSF motif on KNL-1 was sufficient to largely prevent kinetochore localization of endogenous locus-tagged PP1 (PMID: 28698300). Beyond PP1, we have also investigated the PP2A-B56 phosphatase and PP4. In vertebrates PP2A-B56 is recruited to kinetochores by BubR1. The *C. elegans* ortholog of BubR1 (MAD-3), does not localize to kinetochores (PMID: 19109417) and endogenous locus tagging of both B56 subunit-encoding genes (*pptr-1* and *pptr-2*) followed by live imaging did not reveal kinetochore localization of B56 (PMID: 28698300). *In situ* GFP-tagged PP4 also did not localize to kinetochores in this model (PMID: 36719399). Thus, to the best of our knowledge, KNL-1-recruited PP1 is the major kinetochore-localized phosphatase activity in this system. To clarify this point, we have added additional text in the lead-up to the analysis of the KNL-1 PP1-binding mutant.

Based on the (negative) results with the Ndc80 mutant and the KNL-1 PP1-binding mutant, we speculate that dynein removal may be intrinsically triggered by the postulated end-coupled state that orients the chromosome. We base this speculation on the type of data shown in **Fig. 4b-c**, where orientation and dynein removal are tightly coupled (note that the times shown are in seconds). However, direct testing of this speculation is not currently possible as dynein engagement with dynamic microtubule ends is relatively poorly studied and no specific mutants can be engineered that selectively perturb end-coupled interactions versus lattice motility.

2. To my eye, in video 5 and 6 the chromosomes with the dynein-only module show a striking “bounce-back” when the first kinetochore becomes perpendicular to the spindle axis and loses its dynein, and, maybe, again after the second one loses its dynein. I am wondering if this hunch is borne out by careful quantitation. If true, this opens up another possibility: the signal for the removal of kinetochore-only dynein may be the ability of the kinetochore-bound dynein to produce force and processively walk towards the minus end.

We are unclear on what the reviewer means precisely by “bounce-back”. As shown in **Fig. 4b** and the supplemental movies, dynein-mediated orientation is coupled to chromosome positioning in the spindle. To better highlight this point and to address the reviewer's comment, we divided the spindle into 4 quadrants and tracked the position in each quadrant of the 10 chromosomes for which we had simultaneously imaged at high temporal resolution the dynamics of DHC-1::GFP. We present the results of this analysis in **Fig. S3h**, for the orienting

chromosome shown in **Fig. 4b**, and in **Fig. S3i** for the 9 chromosomes whose analysis of orientation and DHC-1::GFP signal is summarized in **Fig. 4c**. In **Fig. S3i**, we plot orientation, position on the spindle and DHC-1::GFP signal on the kinetochore directing orientation). This analysis reveals that orientation is coupled to translocation (as highlighted in the boxed part of **Fig. 4b**) but we do not observe consistent evidence of a “bounce back”.

Regarding the last point, we note that in the dynein-only state, there are a small number of chromosomes that remain persistently trapped in a lateral orientation at the poles (**Fig. 2d** and **Fig. S5c,d**), which is best explained by minus-end directed motility along the microtubule lattice holding them at that spindle position. DHC-1::GFP remains concentrated at the kinetochores of those chromosomes, unlike for the chromosomes that biorient. Thus, we suspect that force production and walking to the minus end are unlikely to trigger dynein removal.

3. Is it feasible to observe chromosome rotation prior to dynein-mediated alignment? Such an observation will also further support the model of chromosome alignment by dynein-plus-end interaction.

In the “blank slate” created by removing all of the major microtubule-directed activities on chromosomes, the chromosomes initially start off in diverse orientations but end up aligning parallel to the spindle axis (**Fig. 1c,e**). Thus, chromosomes can be rotated independently of dynein (or of the other microtubule-directed activities) and we speculate this is achieved by pushing forces from dynamic spindle microtubules. Adding back dynein to this state caused the majority of chromosomes to orient perpendicularly to the spindle axis (**Fig. 2a,c**) – thus, all of the rotation driving this change is dynein-driven. As noted above, this change in angular orientation occurs coincident with translocation towards a spindle pole and is coupled with dynein removal from the kinetochore facing that pole. Thus, all of these events appear to be tightly coupled.

Minor comments:

1. For the wider audience, it will be useful to include a cartoon cross-section of the chromosome to further clarify the localization of the three modules in Figure 1A.

We have added an additional schematic as recommended by the reviewer as well as indicated that the cartoons are corresponding to a side and top view.

2. Figure 1E cartoon: Conventionally, angles are measured counterclockwise. Because the spindle axis is the reference axis, it would be more appropriate to tilt the chromosome the other way and then indicate the angle measurement.

We have modified the schematics as recommended. Please note that the angular measurement reported is the lower angle of the chromosome relative to the pole-to-pole spindle axis and is not measure exclusively in a counter-clockwise manner (e.g. for a tilted chromosome whose counter-clockwise angular measurement is 120°, we report the angle relative to the spindle pole-to-pole axis as being 60°).

3. The percentage scale on many of the chromosome orientation plots changes (e.g., 100% and 80% in Figure 1E). It would ideal to maintain the same scale even if the bars become small.

We have changed all the graphs as recommended.

4. The micrographs and videos also give the impression that the kinetochore-bound dynein is released locally from the side that appears to align first. It would be useful to document this observation because it further supports the notion that the dynein removal cue is dependent on a specific form of dynein-microtubule interaction.

Thank you for pointing this out. We have also had this impression and motivated by the feedback, we include linescan measurements over time for the imaged chromosome in **Fig. 4b** in **Fig. S3g**, which clearly shows asymmetric removal. We also examined the other 9 chromosomes where we have high resolution imaging data, and we observe asymmetric removal in all of them. We have noted this point in the revised text.

REVIEWERS' COMMENTS

Reviewer #1 (Remarks to the Author):

During revision, the authors further improved an already outstanding manuscript. I am delighted to support the manuscript for publication with the highest priority.

Reviewer #2 (Remarks to the Author):

The authors have made a substantial effort for the revised version to address the comments from each of the reviewers, particularly through careful changes to the text and figures. Based on these changes, I find this paper suitable for publication in Nature Communications. This is an excellent paper that makes a strong contribution to understanding the contributions of the different microtubule binding activities at kinetochores.

Although the changes have addressed my prior comments, I do still feel that indicating the experimental system is a valuable part of any abstract and encourage the authors to edit the abstract such that they can add "in *C. elegans* embryos" or other similar wording.

Reviewer #3 (Remarks to the Author):

The authors have appropriately addressed the issues I raised. The manuscript is ready for publication.

RESPONSE TO REVIEWERS

Prevo et al. (NCOMMS-24-12819-A)

Reviewer #1 (Remarks to the Author):

During revision, the authors further improved an already outstanding manuscript. I am delighted to support the manuscript for publication with the highest priority.

We thank the reviewer for their positive assessment of the work.

Reviewer #2 (Remarks to the Author):

The authors have made a substantial effort for the revised version to address the comments from each of the reviewers, particularly through careful changes to the text and figures. Based on these changes, I find this paper suitable for publication in Nature Communications. This is an excellent paper that makes a strong contribution to understanding the contributions of the different microtubule binding activities at kinetochores.

Although the changes have addressed my prior comments, I do still feel that indicating the experimental system is a valuable part of any abstract and encourage the authors to edit the abstract such that they can add "in *C. elegans* embryos" or other similar wording.

We thank the reviewer for their positive assessment of the work and have edited the abstract.

Reviewer #3 (Remarks to the Author):

The authors have appropriately addressed the issues I raised. The manuscript is ready for publication.

We thank the reviewer for their positive assessment of the work.